# Self-Supervised Visual Representation Learning from Hierarchical Grouping

**Xiao Zhang**
University of Chicago
zhang7@uchicago.edu

**Michael Maire**
University of Chicago
mmaire@uchicago.edu

## Abstract

We create a framework for bootstrapping visual representation learning from a primitive visual grouping capability. We operationalize grouping via a contour detector that partitions an image into regions, followed by merging of those regions into a tree hierarchy. A small supervised dataset suffices for training this grouping primitive. Across a large unlabeled dataset, we apply this learned primitive to automatically predict hierarchical region structure. These predictions serve as guidance for self-supervised contrastive feature learning: we task a deep network with producing per-pixel embeddings whose pairwise distances respect the region hierarchy. Experiments demonstrate that our approach can serve as state-of-the-art generic pre-training, benefiting downstream tasks. We additionally explore applications to semantic region search and video-based object instance tracking.

## 1 Introduction

The ability to learn from large unlabeled datasets appears crucial to deploying machine learning techniques across many application domains for which annotating data is too costly or simply infeasible due to scale. For visual data, quantity and collection rate often far outpace ability to annotate, making self-supervised approaches particularly crucial to future advancement of the field.

Recent efforts on self-supervised visual learning fall into several broad camps. Among them, Kingma *et al.* [24] and Donahue *et al.* [12] design general architectures to learn latent feature representations, driven by modeling image distributions. Another group of approaches [16, 10, 28, 35] leverage, as supervision, pseudo-labels automatically generated from hand-designed proxy tasks. Here, the general strategy is to split data examples into two parts and predict one from the other. Alternatively, Wu *et al.* [45] and Zhuang *et al.* [52] learn visual features by asking a deep network to embed the entire training dataset, mapping each image to a location different from others, and relying on this constraint to drive the emergence of a topology that reflects semantics. Across many of these camps, technical improvements to the scale and efficiency of learning further boost results [19, 7, 36, 13]. Section 2 provides a more complete background.

We approach self-supervised learning using a strategy somewhat different from those approaches outlined above. While incorporating aspects of proxy tasks and embedding objectives, a key distinction is that our system's proxy task is itself generated by a (simpler) trained vision system. We thus seek to bootstrap visual representation learning in a manner loosely inspired by, though certainly not accurately mirroring, a progression of simple to complex biological vision systems. This is an under-explored, though not unrecognized, strategy in computer vision. Serving as a noteworthy example is Li *et al.*'s [30] approach of using motion as a readily available, automatically-derived, supervisory signal for learning to detect contours in static images. We focus on the next logical stage in such a bootstrapping sequence: using a pre-existing contour detector to automatically define a task objective for learning semantic visual representations. Figure 1 illustrates how this primitive

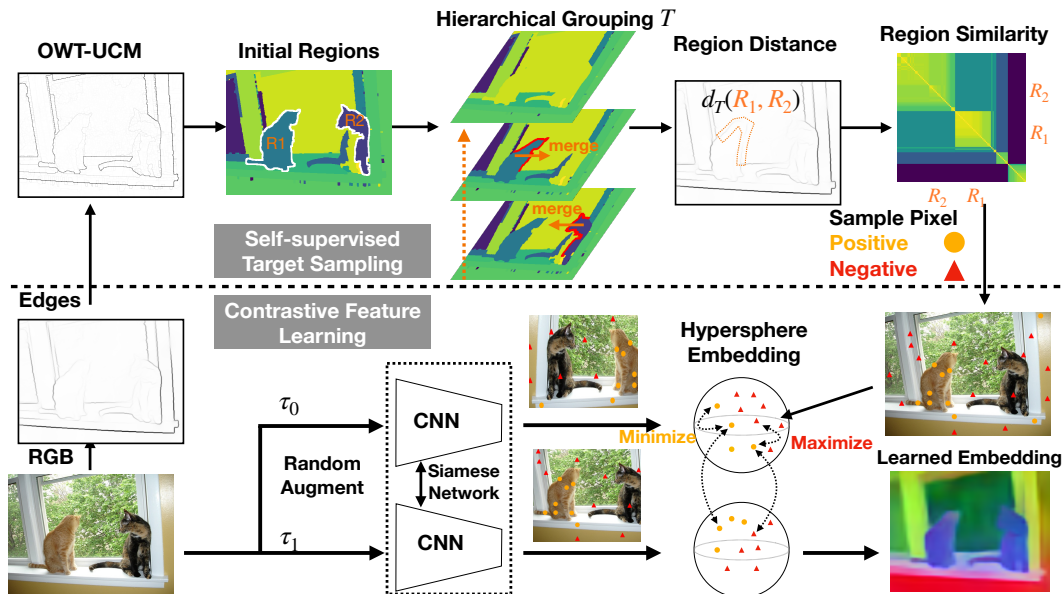

Figure 1: **Bootstrapping semantic representation learning via primitive hierarchical grouping.** *Top: Self-Supervised Target Sampling.* From a hierarchical segmentation of an image (*i.e.,* a region tree $T$), rendered here as a boundary strength map (OWT-UCM [2]), we define distance between regions $d_T(R_1, R_2)$ according to the level in the hierarchy at which they merge. Treating this distance as a similarity measure between pixels, we sample positive and negative pairs of pixels, according to their grouping likelihood in the hierarchy. *Bottom: Contrastive Feature Learning.* A contour detector [2], trained on a small dataset [34], produces hierarchical segmentations across a larger unlabeled image set. Automatically extracted positive and negative pixel pairs serve to drive a from-scratch initialized CNN to learn to predict pixel-wise embeddings which respect the region hierarchy. Unlike SegSort [21], our pipeline does not merely fine-tune ImageNet [9] pre-trained models for semantic segmentation, but instead addresses representation learning entirely from scratch.

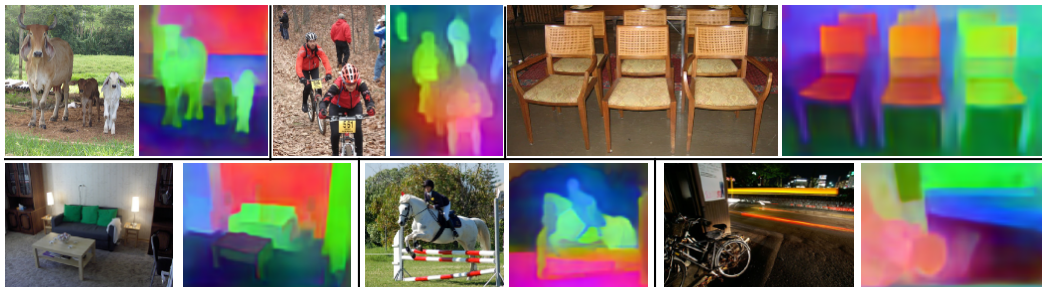

Figure 2: **Visualization of feature embeddings.** We apply PCA to the embeddings produced by a CNN trained using the self-supervised bootstrapping approach of Figure 1. On validation examples, we display the first three PCA components as an RGB image. The output feature representations capture aspects of semantic category (*e.g.,* top left) and object instance identity (*e.g.,* top right).

visual system, combined with a modern contrastive feature learning framework, trains a convolutional neural network (CNN) to produce semantic embeddings (Figure 2). We defer full details to Section 3.

Our system not only leverages contours to learn visual semantics, but also leverages a small amount of annotated data to learn from a vastly larger pool of unlabeled data. Our visual primitive of contour detection is trained in a supervised manner from only 500 annotated images [34]. This primitive then drives self-supervised learning across datasets ranging from tens of thousands to millions of images; in this latter phase, our system does not utilize any annotations and trains from randomly initialized parameters. This is a crucial distinction from SegSort [21], whose pipeline bears coarse resemblance to our Figure 1. SegSort's "unsupervised" learning setting still relies on starting from ImageNet

pre-trained CNNs; its "unsupervised" aspect is only with respect to forgoing use of segmentation ground-truth. In contrast, we address the problem of representation learning entirely from scratch, save for the 500 annotated images of the Berkeley Segmentation Dataset [34].

ImageNet, even without labels, is curated: most ImageNet images contain a single category. This provides some implicit supervision, which may bias the self-supervised work that experimentally targets learning from ImageNet, including MoCo [19], InstFeat [48] and others [45, 7]. Many use cases for self-supervision will lack such curation. As our bootstrapping strategy utilizes a visual primitive geared toward partitioning complex scenes into meaningful components, it is a better fit to learning on unlabeled examples from datasets containing scenes (*e.g.,* PASCAL [14], COCO [31]).

Using a similar siamese network, we outperform InstFeat [48] by a large margin on the task of learning transferable representations from PASCAL and COCO images alone (disregarding labels). In this setting, our results are competitive with those of the state-of-the-art MoCo system [19], while our method remains simpler. We do not rely on a momentum encoder or memory bank. Even with this simpler training architecture, our segmentation-aware approach enhances the efficiency of learning from complex scenes. Here, our pre-training converges in under half the epochs needed by MoCo to learn representations with comparable transfer performance on downstream tasks.

In addition to evaluating learned representation quality on standard classification tasks, Sections 4 and 5 explore applications to semantic region search and instance tracking in video. Using similarity in our learned feature space to conduct matching across images and frames, we outperform competing methods in both applications. Our results point to a promising new pathway of crafting self-supervised learning strategies around bootstrapping the training of one visual module from another.

## 2   Related Work

**Self-Supervised Representation Learning.** Approaches to self-supervised visual learning that train networks to predict one aspect of the data from another have utilized a variety of proxy tasks, including prediction of context [38], colorization [50, 29], cross-channels [51], optical-flow [49], and rotation [16]. Oord *et al.* [36] train to predict future representations in the latent space of an autoregressive model. As objects move coherently in video, Mahendran *et al.* [33] learn pixel-wise embeddings for static frames, such that their pair-wise similarity mirrors that of optical flow vectors.

Another family of methods casts representation learning in terms of clustering or embedding objectives. DeepCluster [5] iterates between clustering CNN output representations to define target classes and re-training the CNN to better predict those targets. Contrastive multiview coding [42] sets the objective as mapping different views of the same scene to a common embedding location, distinct from other scenes. Ye *et al.* [48] apply similar intuition with respect to data augmentation. Instance discrimination [45] formulates feature learning as a non-parametric softmax prediction problem, enforcing consistency between a predicted hypersphere embedding and a counterpart maintained in memory banks. Following Wu *et al.* [45], Zhuang *et al.*'s local aggregation approach [52] uses additional clustering steps to reason about embedding targets.

Differing from the siamese network of Ye *et al.* [48] and the dataset-level memory bank of Wu *et al.* [45], momentum contrast (MoCo) [19] uses a moving-average encoder for reference embeddings. This offers scalability superior to a memory bank. Self-supervised networks trained with MoCo outperform their ImageNet-supervised counterparts, as benchmarked by fine-tuning to multiple tasks.

Like MoCo, our approach also benefits from a feature learning paradigm that jointly considers augmentation invariance and negative example sampling. But, instead of learning feature representation only at image level, our method learns pixel-wise semantic affinity in the context of regions. Our system relies on a simpler siamese architecture, rather than requiring a moving-average encoder.

**Image Segmentation.** The classic notion of image segmentation – partitioning into meaningful regions without necessarily labeling according to known semantic classes – has a rich history. Given the duality between regions and their bounding contours, modern approaches often focus on the problem of contour detection. Though deep neural networks are now the dominant tool for contour detection [41, 4, 46, 25, 47], the best prior approaches [2, 3, 11] deliver somewhat respectable results.

Pre-training CNNs on ImageNet before fine-tuning them for contour detection provides accuracy gains [46]. But, as our purpose is to bootstrap representation learning from contours, obviating the

need for ImageNet supervision, we do not want ImageNet labels used in our contour detector training pipeline. For experiments, we therefore select an older detector, based on random forests [11], along with traditional machinery (OWT-UCM) [2] for converting contours into region hierarchies. Both components are trained using only the 500 labeled images of the BSDS [34].

**Representation Learning using Segmentation.** Several works incorporate segmentation into representation learning. Fathi *et al.* [15] formulate the task of instance segmentation as metric learning. They train a network for instance-aware embedding by optimizing feature similarity of pixels sampled within or cross-instances. Kong *et al.* [27], adopting a similar objective, combine a CNN with a differentiable mean-shift clustering approach to learn instance segmentation. Chen *et al.* [8] address video instance segmentation via pixel embedding with a modified triplet loss. Pathak *et al.* [37] learn representations for recognition tasks by predict moving object segmentation from static frames.

SegSort [21] utilizes an iterative grouping algorithm in EM (expectation maximization) fashion, learning a segmentation-aware embedding of pixels onto a hypersphere. Specifically, it leverages regions (computed by OWT-UCM [2] on HED [46] contours) as defining separation criteria, maximizing pairwise intra-similarity and inter-contrast for the pixels within the same or different regions, respectively. Unlike supervised counterparts, SegSort can learn semantic segmentation on top of ImageNet [9] pre-trained CNNs. In this mode, contours (as opposed to ground-truth per-pixel class labels) provide the only additional supervisory signal for learning region semantics.

While our work shares a similar spirit with SegSort [21], we aim to bootstrap learning of semantic region representations entirely from scratch, removing the dependence on ImageNet pre-training for both the primary CNN and the contour detection component.

## 3 Bootstrapping Semantics from Grouping

We train a convolutional neural network $\phi(\cdot)$, which maps an input image $\mathbf{I}$ into a spatially-extended feature representation $\mathbf{F} = \phi(\mathbf{I})$. Let $\mathbf{F}(i) \in \mathbb{R}^d$ denote the output $d$-dimensional feature embedding of pixel $i$. We adopt a contrastive learning objective that operates on a pixel level. Defining $\text{sim}(\mathbf{F}(i), \mathbf{F}(j)) = \mathbf{F}(i)^T \mathbf{F}(j) / (||\mathbf{F}(i)|| \cdot ||\mathbf{F}(j)||)$ as the cosine similarity between feature vectors $\mathbf{F}(i)$ and $\mathbf{F}(j)$, we want to learn optimal network parameters $\phi^*$ as follows:

$$\phi^* = \arg\max_{\phi} \sum_i \sum_{m \in Pos(i)} \frac{\exp(\text{sim}(\mathbf{F}(i), \mathbf{F}(m)))}{\exp(\text{sim}(\mathbf{F}(i), \mathbf{F}(m))) + \sum_{n \in Neg(i)} \exp(\text{sim}(\mathbf{F}(i), \mathbf{F}(n)))} \quad (1)$$

where $Pos(i)$, $Neg(i)$ denote the pixels in the same or different semantic categories as pixel $i$.

Unlike the supervised setting, where annotations determine $Pos(i)$, $Neg(i)$, we must automate estimation of these relationships. In designing such a procedure, we operate under stringent assumptions: (1) the network is initialized from scratch, and (2) training images may contain complex scenes.

**Grouping Primitive.** We deriving guidance from a visual grouping primitive to sample $Pos(i)$ and $Neg(i)$. Specifically, we adopt a contour detector $\phi_E$, which, acting on image $I$, produces an edge strength map $E = \phi_E(I)$. We then convert $E$ into a region map $R$ using a hierarchical merging process which repeatedly removes the weakest edge separating two regions. The real-valued edge strength at which two distinct regions merge defines a distance metric, which we extend to a notion of distance between pixels. This is precisely the procedure for constructing an ultrametric contour map (UCM) [1], which can equivalently be regarded as as both a reweighted edge map, and a tree $T$ defining a region hierarchy. The leaves of $T$ are the initial, finest-scale, regions; interior nodes represent larger regions formed by merging their children and have a real-valued height in the hierarchy equal to the distance between their child regions. Figure 1 (top, center) shows an example.

As remarked upon earlier, we utilize the structured forest edge detector of Dollár and Zitnick [11], trained on a small dataset (BSDS [34]). In constructing the region tree, we follow Arbeláez *et al.* [2], applying their variant of the oriented watershed transform (OWT) prior to computing the UCM.

**Pixel Sampling from Region Metric.** Denote the distance between regions $R_1$ and $R_2$ defined by the UCM-derived region tree $T$ as $d_T(R_1, R_2)$. We lift this region distance to serve as a measure of the probability that two pixels belong in the same semantic region. Considering two pixels $i, j$, we express $P(j \in Pos(i))$, the probability to assign the pixel $j$ into the positive set of $i$, as a function of

region distance. For $i \in R_a, j \in R_b$:

$$P(j \in Pos(i)) = \frac{\exp(-d_T(R_a, R_b)/\sigma_p)}{\sum_{m \neq a}^{M} \exp(-d_T(R_a, R_m)/\sigma_p) + 1} \quad (2)$$

where $\sigma_p$ is a temperature parameter to control the concentration on region distance, and we manually cast the self-similarity to one: $\exp(-d_T(R_a, R_a)/\sigma_p) \rightarrow 1$ to further concentrate positive sampling within the anchor regions. In experiments, we find this trick leads to better positive sampling and performance. Similarly, we can define the probability of assigning $j$ to $i$'s negative set as:

$$P(j \in Neg(i)) = \frac{\exp(d_T(R_a, R_b)/\sigma_p)}{\sum_{m \neq a}^{M} \exp(d_T(R_a, R_m)/\sigma_p)} \quad (3)$$

Note that we do not formulate $P(j \in Neg(i))$ as the complement of $P(j \in Pos(i))$, for the purpose of excluding the pixels whose assignments are vague. With these defined distributions, we can sample corresponding reference pixels from the image.

**Augmented Reference Sampling.** Asking vision systems to produce feature representations that are invariant to common image transformations (*e.g.,* color change, object deformation, occlusion) appears to be extremely helpful in learning semantic features. To this end, we augment our sampling of positive and negative pixel pairs in two ways:

- *Image Augmentation.* We impose data augmentation on training images to collect extra positive and negative pairs. Specifically, suppose we have sampled $Pos(i)$ and $Neg(i)$ for pixel $i$, and augmented input image with transformation $\tau$. We can then have augmented pixels pairs $Pos(i)_{aug} = \tau(Pos(i))$ and $Neg(i)_{aug} = \tau(Neg(i)))$ to sample referenced features on $\phi(\tau(\mathbf{I}))$.
- *Crossing-image Negative Sampling (cns).* Capturing object-level, rather than merely instance-level, semantics requires considering relationships across the entire dataset; we need a signal relating reference pixels from different images. We adopt an approach similar to [45, 19, 44] and randomly sample negative features from the dataset (but operationalized on a per-pixel level). We denote these randomly sampled negatives as $Neg(i)_{cns}$ for selected anchor pixel $i$.

**Optimization.** By jointly considering all sources to provide reference feature vectors, we can modify our target in Equation (1) by updating: $Pos(i)$ with $Pos^+(i) = Pos(i) \cup Pos(i)_{aug}$ and $Neg(i)$ with $Neg^+(i) = Neg(i) \cup Neg(i)_{aug} \cup Neg(i)_{cns}$. During training, we end-to-end optimize Equation (1) starting from a randomly initialized network $\phi$, as to produce feature vectors $\mathbf{F}(i)$ which encode fine-grained pixel-wise semantic representations.

# 4 Experiment Settings

## 4.1 Datasets and Preprocessing

**Datasets.** We experiment on datasets of complex scenes, with variable numbers of object instances: PASCAL [14] and COCO [31]. PASCAL provides 1464 and 1449 pixel-wise annotated images for training and validation, respectively. Hariharan *et al.* [17] extend this set with extra annotations, yielding 10582 training images, denoted as *train_aug*. The COCO-2014 [31] dataset provides instance and semantic segmentations for 81 foreground object classes on over 80K training images. In unsupervised experiments, we train the network on the union of images in the *train_aug* set of PASCAL and *train* set of COCO. We evaluate learned embeddings on the PASCAL *val* set by training a pixel-wise classifier for semantic segmentation on PASCAL *train_aug*, set atop frozen features.

We also benchmark learned embeddings on the DAVIS-2017 [40] dataset for the task of instance mask tracking. We *ONLY* train on PASCAL *train_aug* and COCO *train*, without including any images from DAVIS-2017. After self-supervised training, we directly evaluate on the *val* set of DAVIS-2017 as to propagate initial instance masks to consecutive frames through embedding similarity.

**Edges and Regions.** Our proposed self-supervised learning framework starts with an edge predictor. We avoid neural network based edge predictors like HED [46], which depend upon an ImageNet pre-trained backbone. We instead turn to structured edges (SE) [11], which only leverages the small supervised BSDS [34] for training. From SE-produced edges, we additionally compute a spectral edge signal (following [2]), and feed a combination of both edge maps into the processing steps of OWT-UCM [2]. This converts the predicted edges a region tree $T$, which we constrain to have no more than 40 regions at its finest level.

## 4.2 Implementation Details

**Network Design.** We use a randomly initialized ResNet-50 [20] backbone. To produce high fidelity semantic features, we adopt a hypercolumn design [18] that combines feature maps coming from different blocks. We keep the stride of ResNet-50 unchanged and adopt a single $3 \times 3$ Conv-BN-ReLU block to project the last outputs of the Res3 and Res4 blocks into 256-channel feature maps. These two feature maps are both interpolated to match the spatial resolution of a $4\times$ downsampled input image before concatenation. Finally, we project the resulting feature map using a single $3 \times 3$ convolution layer, to produce a final 32-channel feature map as our output semantic embedding $\mathbf{F}$.

**Training.** We use Adam [23] to train our model for 80 epochs with batch size 70. We initialize learning rate as 1e-2 which is then decayed by 0.1 at 25, 45, 60 epochs, respectively. We perform data augmentation including random resized cropping, random horizontal flipping, and color jittering on input images, which are then resized to $224 \times 224$ before being fed into the network. For one image, we randomly sample 7 regions and, for each region, sample 10 positive pixels and 5 negative pixels. We use $\sigma_p = 0.8$ for all experiments.

In experiments fine-tuning on PASCAL $train\_aug$, we freeze all the base features and only update parameters of a newly added head, the Atrous Spatial Pyramid Pooling (ASPP) module from DeepLab V3 [6]. Here, we use SGD with weight decay $5e^{-4}$ and momentum 0.9 to optimize the pixel-wise cross entropy loss for 20K iterations with batch size 20. We randomly crop and resize images to 384x384 patches. The learning rate starts at 0.03 and decays by 0.1 at 10K and 15K iterations.

# 5 Results and Analysis

## 5.1 Evaluation of Learned Self-Supervised Representations

To compare with other representation learning approaches: InstFeat [48], MoCo [19], and Seg-Sort [21], we use the official code released by the respective authors to train a ResNet-50 backbone on the same unified dataset of COCO $train$ and PASCAL $train\_aug$ set. We preserve the default parameters of these approaches. For InstFeat [48] and SegSort [21], we set the total training epochs to 80 and rescale the timing of learning rate decay accordingly. We find 80 epochs is sufficient for convergence of these models.

For SegSort [21], we replace their network, which is built upon an ImageNet pre-trained ResNet-101, with a ResNet-50 that is initialized from scratch. We update the related parameters, *e.g.,* learning rate, decayed epochs, total epochs, and region map to be consistent with our setting. This allows us to compare with SegSort as a truly unsupervised approach, rather than one that relies on ImageNet pre-training. Table 1 reports results after fine-tuning for PASCAL semantic segmentation.

Training with a similar siamese architecture on PASCAL+COCO, our method outperforms Inst-Feat [48] by a large margin (46.51 *vs.* 38.11 mIoU). We outperform SegSort [21], which neither leverages information from the region hierarchy nor samples negatives in a cross-image fashion, by a larger margin (46.51 *vs.* 36.15).

Though equipped with neither a momentum encoder nor a memory bank, our method achieves comparable results to a variant of MoCo [19] (46.51 *vs.* 47.01 mIoU), while requiring far fewer training epochs (80 *vs.* 200). Restricted to only 80 pre-training epochs, we significantly outperform MoCo (46.51 *vs.* 42.07). Given less pre-training data (PASCAL only, Table 1 bottom), we similarly observe a substantial advantage over MoCo (43.51 *vs.* 32.04 mIoU). Together, faster convergence and superior performance with less data suggest that our ability to exploit regions drastically improves the efficiency of unsupervised visual representation learning.

This efficiency hypothesis is further supported by the fact that our method continues to improve if it can take advantage of higher quality edges. A variant built using from the advanced edge detector HED [46] reaches 48.82 mIoU, outperforming all competing approaches. Note, however, that HED itself utilizes ImageNet pre-training, so this is an ablation-style comparison that does not obey the same strict restriction, of only relying on unlabeled PASCAL+COCO images, as all other entries in the corresponding section of Table 1.

Besides freezing the backbone, we also run an end-to-end fine-tuning experiment on a PAS-CAL+COCO pre-trained backbone, following the evaluation protocol in He *et al.* [19]. Our method

| Method | Cross-Image Negative Sample | Unsupervised Pre-train | Pre-train Epochs | mIoU |
|---|---|---|---|---|
| Random Init | None | None | - | 12.99 |
| Random Init(End to end) | None | None | - | 45.43 |
| ImageNet Pre-train | None | None | - | 70.54 |
| MoCo [19] | Image | ImageNet | 200 | 65.53 |
| **Ours** | Pixel(cns=5) | ImageNet | 200 | 51.02 |
| **Ours**[†] | Pixel(cns=5) + Image | ImageNet | 200 | 64.70 |
| InstFeat [48] | Image | PASCAL + COCO | 80 | 38.11 |
| SegSort [21] | None | PASCAL + COCO | 80 | 36.15 |
| MoCo [19] | Image | PASCAL + COCO | 80 | 42.07 |
| MoCo [19] | Image | PASCAL + COCO | 200 | 47.01 |
| **Ours** | Pixel(cns=5,20) | PASCAL + COCO | 80 | 46.51 |
| **Ours**(HED) | Pixel(cns=5,20) | PASCAL + COCO | 80 | 48.82 |
| MoCo [19] | Image | PASCAL | 400 | 32.04 |
| Ours | Pixel(cns=5) | PASCAL | 400 | 43.51 |
| Ours | Pixel(cns=5) | PASCAL + COCO $\times$ 0.5 | 140 | 44.92 |
| Ours | Pixel(cns=5) | PASCAL + COCO | 80 | 45.57 |
| Ours | None | PASCAL + COCO | 80 | 39.63 |

Table 1: **Quantitative evaluation of learned self-supervised representations.** We train classifiers on top of frozen features to predict semantic segmentation, and benchmark on the PASCAL *val* set. When using complex images (PASCAL + COCO) for unsupervised pre-training *(bottom half of table)*, our approach outperforms [48, 21]. Moreover, without using a momentum encoder or memory bank, we exceed MoCo [19] on training efficiency (better mIoU at 80 epochs), while achieving accuracy comparable to a much longer-trained MoCo.

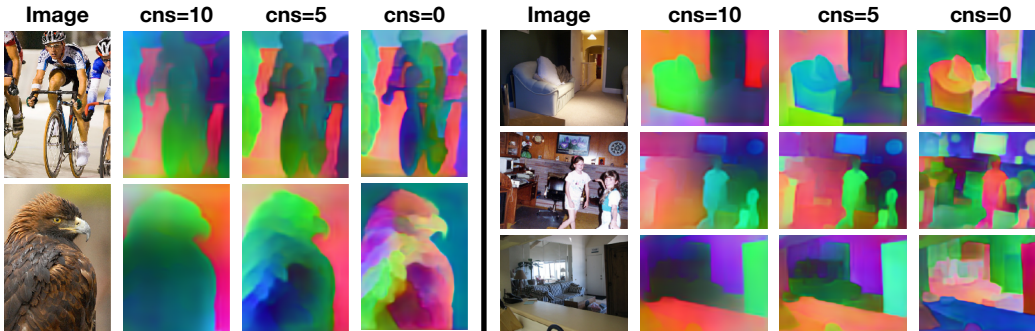

Figure 3: **Effect of crossing-image negative sampling.** Notation $cns = m$ denotes, for each anchor pixel, we randomly sample $m$ extra features as negative references. Adjusting $cns$ effectively trades-off between learned embeddings that reflect semantic categories *vs.* local instance boundaries.

achieves comparable results with MoCo [19] (47.2 *vs.* 46.9 mIoU) when both are trained for 80 epochs. Here, MoCo [19] reaches 55.0 when trained for 200 epochs.

We perform ablations on cross-image negative sampling ($cns$), reported in Table 1 (bottom). Figure 3 shows that increasing $cns$ yields feature embeddings more consistent with semantic partitioning.

We also experiment with unsupervised training over ImageNet, where MoCo performs well under the implicit assumption that most unlabeled images only contain only a single category. Simply adding our pixel-wise contrastive target on top of MoCo outperforms the MoCo baseline when benchmarked at 30 epochs (55.13 *vs.* 53.84), but, as shown in Table 1 (top, Ours[†]), we do not witness relative improvement at 200 epochs. However, this does suggest that region awareness can help to boost unsupervised learning efficiency, even on ImageNet.

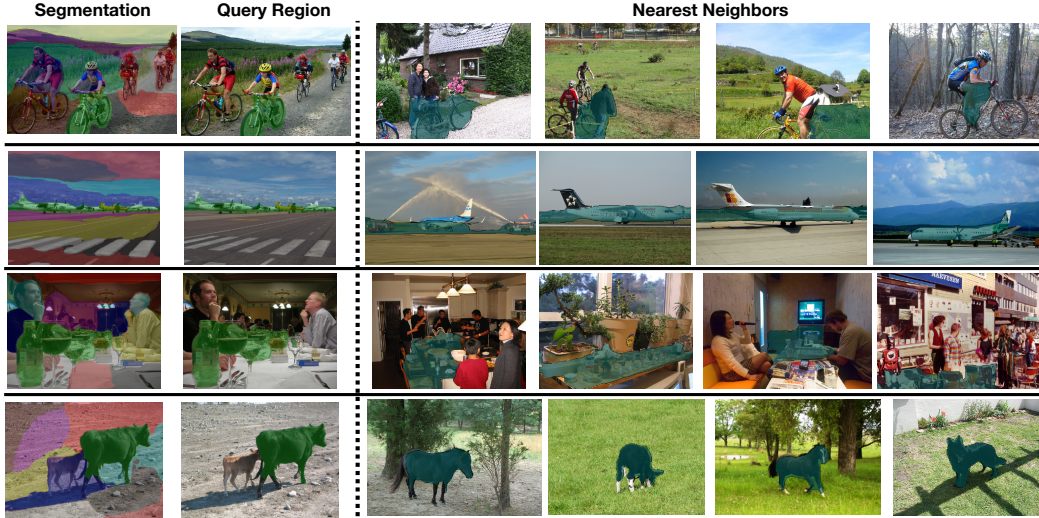

**Segmentation**  **Query Region**  **Nearest Neighbors**

Figure 4: **Qualitative evaluation of segment search.** We leverage our learned embeddings to partition images via K-means clustering and serve as region descriptors for nearest neighbor search.

| | Mean | Bus | Aero | Car | Person | Cat | Cow | Bottle |
|---|---|---|---|---|---|---|---|---|
| Class Frequency | - | 4.01% | 3.00% | 6.52% | 25.41% | 10.66% | 1.81% | 1.70% |
| SegSort [21] | 10.17 | 9.36 | 22.28 | 11.24 | 30.28 | 17.51 | 0.31 | 0.00 |
| Ours | **24.60** | 50.00 | 47.41 | 41.89 | 49.41 | 36.00 | 3.15 | 8.46 |

Table 2: **Quantitative evaluation of segment search.** We report mean and per-class IoU (for selected classes) on PASCAL *val*. Our approach works best on high frequency or rigid object classes.

## 5.2   Semantic Segment Retrieval

We also adopt a direct approach to examine our learned embeddings. We first partition the images of PASCAL $train\_aug$ and $val$ set into a fixed number of segments by running K-means clustering on the embedding. Then we use a region feature descriptor, computed by averaging the feature vectors over all pixels within a segment, to retrieve nearest neighbors of the $val$ set regions from the $train\_aug$ set. We report qualitative (K=10, Figure 4) and quantitative (K=15, Table 2) results. Without any supervised fine-tuning, our learned representations reflect semantic categories and object shape.

## 5.3   Instance Mask Tracking

In this task, we track instance masks by fetching cross-frame neighboring pixels measured under feature similarity induced by our output embedding. Specifically, we predict the instance class of pixel $i$ at time step $t$ by $y_t(i) = \sum_k \sum_j \mathrm{sim}(\mathbf{F}_t(i), \mathbf{F}_{t-k}(j)) y_{t-k}(j)$, where $\mathbf{F}_i(t)$ denotes the feature vector of pixel $i$ at time step $t$, which we also augmented with spatial coordinates $i$. We utilize the $k$ previous frames to propagate labels. $y_t(i)$ is a one-hot vector indicating the instance class assignment for pixel $i$ at time frame $t$. In our experiment, we choose $k = 2$ and $j$ as the set of 10 nearest neighbors of pixel $i$.

We evaluate performance using region similarity $\mathcal{J}$ and contour accuracy $\mathcal{F}$ (as defined by [39]), with Table 3 reporting results. Our feature representation, learned by respecting the region hierarchy, benefits temporal matching for difficult cases, *e.g.,* large motion, where local intensity is not reliable. Though not optimized for precise temporal matching, our approach still outperforms recent state-of-the-art video-based unsupervised approaches in both region and boundary quality benchmarks.

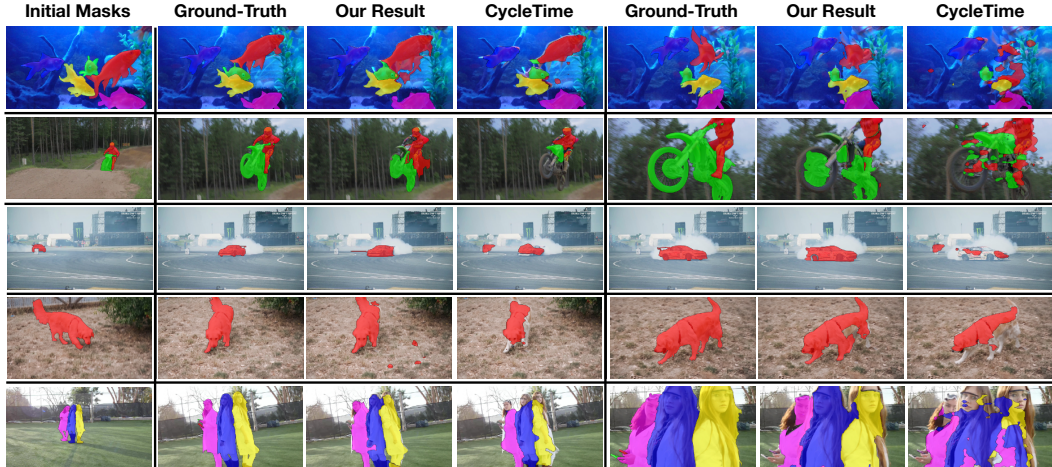

| Initial Masks | Ground-Truth | Our Result | CycleTime | Ground-Truth | Our Result | CycleTime |

Figure 5: **Video-based instance tracking on DAVIS-2017 [40].** Our method outperforms video-based unsupervised approach CycleTime [44] on both region and boundary quality.

| Method | $\mathcal{J}$(Mean)↑ | $\mathcal{F}$(Mean)↑ | Method | $\mathcal{J}$(Mean)↑ | $\mathcal{F}$(Mean)↑ |
|---|---|---|---|---|---|
| Identity | 22.1 | 23.6 | Random Init | 27.7 | 26.6 |
| FlowNet2 [22] | 26.7 | 25.2 | SIFT Flow [32] | 33.0 | 35.0 |
| DeepCluster [5] | 37.5 | 33.2 | Video Colorization [43] | 34.6 | 32.7 |
| CycleTime [44] | 41.9 | 39.4 | mgPFF [26] | 42.2 | 46.9 |
| Ours ($cns = 0$) | 43.1 | 46.7 | Ours ($cns = 5$) | **47.1** | **48.9** |

Table 3: **Quantitative evaluation of instance mask tracking on DAVIS-2017 [40].** We benchmark region quality ($\mathcal{J}$) and boundary quality ($\mathcal{F}$) on the validation set, using the respective metrics defined by Perazzi *et al.* [39]. Our method outperforms competitors on both metrics.

## 6 Conclusion

We propose a self-supervised learning framework, only leveraging a visual primitive predictor trained on a small dataset, that bootstraps visual feature representation learning on large-scale unlabeled image sets by optimizing a pixel-wise contrastive loss to respect a primitive grouping hierarchy. We demonstrate the effectiveness of our pixel-wise learning target as deployed on unlabeled images of complex scenes with multiple objects, through fine-tuning to predict semantic segmentation. We also show that our learned features can directly benefit the downstream applications of segment search and instance mask tracking.

## 7 Broader Impact

As an advance in self-supervised visual representation learning, our work may serve as a technical approach for a wide variety of applications that learn from unlabeled image datasets, with impact as varied as the potential applications. We believe that a compelling and practical use case is likely to be in domains where human annotation is especially difficult, such as medical imaging, and are hopeful that that further development of these techniques will eventually have a positive impact in medical and scientific domains.

## Acknowledgments and Disclosure of Funding

The authors have no competing interests.

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
