[Reviews · NeurIPS 2020]

Review 1

Summary and Contributions: === Post rebuttal update begins === I thank the authors for addressing some of my concerns. I, however, disagree with several of the arguments put forward in the rebuttal. I have nevertheless updated my overall score as the authors provided/promised some of the requested experiments. I detail my concerns below: "aim of Self-supervised learning is to create universal visual representations": A substantial part of the community, however, is still interested in transferable representations. This pursuit is valuable in its own way as it tries to generalize to very novel settings with very limited data. For example, we may not hope for universal representations that work on medical images, satellite images, faces, textures, game data etc. all at once, but we can certainly hope for representations that transfer well to many domains with only a few labeled example. In fact, 1% and 10% ImageNet end-to-end fine-tuning is slowly becoming more common. The pursuit of universal representations is also of interest to the NeurIPS community. I request that this goal be stated clearly in the final manuscript. To propose a new method and to *only* evaluate it in a new setting is incomplete. One must evaluate on both old and new settings and include several baselines in the new setting along with substantial arguments in favour of it. Despite these disagreements the authors present fine-tuning results with COCO + VOC and agree to include full 200 epoch ImageNet results. Based on this I have upgraded my overall score. === Post rebuttal update ends === This paper propose a method to learn per-pixel neural representations using hierarchical segmentation as the supervisory signal. The segmentation model is itself learned using only 500 images manually annotated with object boundaries. Given predicted edge-maps the authors generate hierarchically grouped regions based on prior work. An inter-region distance is defined based on this output. This distance defines the probability of a pixel being considered as positive w.r.t. another anchor pixel and similarly the probability of a pixel being considered as negative. Pixels sampled according to these distributions are used to train per-pixel neural representations using a contrastive loss. For evaluation the representations are frozen and used as input to a segmentation "head" for the Pascal VOC semantic segmentation benchmark. The representation is also evaluated directly by treating it is as a pixel embedding and apply clustering and to generate regions and using them for segment search and video mask tracking.

Strengths: This paper further's the idea of using low level vision pipelines to create proxy tasks for self-supervised learning. This is relevant to the NeurIPS community. The experimental analysis examines several aspects of the learned representations; for downstream semantic segmentation, for unsupervised segmentation and segment search, and video tracking. It carefully ablates the availability of more self-supervision data, and the use of negative pixels from other images. The use of negative samples from other images induces semantics. This is illustrated qualitatively in figure 3 and suggested quantitatively in Table 1. Some of the prior works that did per-pixel self-supervision, in the context of images, did not include such negatives.

Weaknesses: Experiments in Table 1 deviate significantly from the protocols used in prior work [19, 5, 10 for example]. There are two main differences: 1. The representation is frozen when evaluating downstream for semantic segmentation. Thus the representation is evaluated as opposed to its transferrability. Frozen representations were only used in prior work for linear evaluation on ImageNet and Places205 [19, 5 for example]. For segmentation, on the other hand, the model is always fine-tuned end-to-end. End-to-end fine-tuning is important here because these prior works learned per-image embeddings as opposed to per-pixel embeddings, whereas the downstream task is per-pixel semantic segmentation. Thus the comparison against [19] in row 10-11 is unfair to [19]. I also note that the 65.53% mIoU reported for [19] is less than 72.5% reported in Table 6 of [19]. I presume this is because of freezing the representation (the segmentation head is also different but the ASPP head used in this submission might be more powerful). Note that this is about prior work's embeddings being image level as opposed to having a single object in each image for self-supervision. The current evaluation protocol confounds these factors. 2. Pre-training on COCO + Pascal as opposed to ImageNet. The authors argue that this simulates a more realistic scenario for self-supervision where data is not curated to contain dominant single objects. While I agree with this intuition, it hurts the analysis in two ways: (a) Are Pascal and COCO less curated? A discussion about this will help make the argument stronger. They may contain more objects per image but the question is more about whether they have many images of unrelated object classes and whether the data distribution is long tailed. [19] tried to simulate an un-curated setting using the Instagram-1B dataset. (b) Future work will try to compare against the large body of prior work which pre-trained on ImageNet. It will be difficult to compare against this submission because the ImageNet pre-trained results are preliminary (only 30 epochs of training). I suggest that the authors complement Table 1 with results in more standard settings, or argue more strongly as to why their protocol is better. In the latter case, they should incorporate more baselines in the table including some that train per-pixel representations.

Correctness: Minor errors only Line 221 quotes 36.15 whereas Table 1 row 9 quotes 34.74 Line 224-226: Please highlight that these results are preliminary as these models were trained for only 30 epochs on ImageNet. Please clarify whether figures 3, 4, 5 are a random selection. Please also show some failure modes in the qualitative results.

Clarity: The paper is well written overall. The following points could be explained in more detail. How does equation (2) form a distribution that adds to 1. \sum_j P(j \in Pos(i)) = 1? What is P(j \in Neg(i)_{cns}) ? The authors describe it is similar to [42, 19, 41]. Do you use the queue based dictionary of [19]? Pseudo code for the construction of d_T(R_i, R_j) and for the sampling will be great. Also how does eqn (1) change when incorporating image augmentation and cross-image negative sampling? Do you compute representations both from the original image and the augmented one?

Relation to Prior Work: The related work covers several papers. I recommend adding the following to make it more comprehensive. [M1] Pathak, Deepak, et al. "Learning features by watching objects move." CVPR 2017. This paper extracts unsupervised foreground-background segments for video frames and defines a proxy task by using these segments as ground truth. It overlaps with the current submission in the use of segmentation as a supervisory signal. This hard assignment was relaxed into a soft kernel matching objective by [M2] A Mahendran, J Thewlis, A Vedaldi, "Cross pixel optical-flow similarity for self-supervised learning" ACCV 2018. [M3] Xiaohang Zhan, Xingang Pan, Ziwei Liu, Dahua Lin, Chen Change Loy "Self-Supervised Learning via Conditional Motion Propagation" CVPR 2019. This paper uses optical flow to define the proxy task. It overlaps in the use of low-level vision to bootstrap self-supervision. They also learn per-pixel representations and evaluate downstream on semantic segmentation. Infact some of these should ideally be included in Table 1, but I understand that the protocols used differ considerably from [19] and so comparing against all of them is difficult. Contrastive learning (CPC style) was used in a pixel wise setting by [M4] Tengda Han and Weidi Xie and Andrew Zisserman, "Video Representation Learning by Dense Predictive Coding", ICCV workshops 2019.

Reproducibility: Yes

Additional Feedback: A lot of details are provided to aid reproducibility, but some more equations as discussed above with make it easier. Regarding broader impact: I request to authors to discuss both positive and negative outcomes.


Review 2

Summary and Contributions: The paper proposes a self-supervised representation learning approach for imaging data using a pixel-wise contrastive learning objective. Distances between pixel representations are obtained by leveraging a hierarchical region structure. The key contribution is a visual representation learning approach that avoids pre-training on large-scale datasets such as ImageNet.

Strengths: Main strength of the paper is a novel approach for annotation-efficient self-supervised training. Only the utilized contour detection method requires some densely annotated training data. Another strength is the extensive evaluation on different tasks (semantic search and region tracking) and comparison to several baselines and state-of-the-art methods.

Weaknesses: A weakness of the paper is that the evaluation part is a bit light on the discussion. The paper mostly presents the raw numbers without providing much of an analysis of the results.There is no mentioning of failure cases or limitations. Generally, I found the second part of the paper feels a bit rushed and some important details are missing about the experimental settings. For example, it is unclear to me what the entry 'Image' means under column 'Cross-Image Negative Sample' in Table 1. In that caption, it is also mentioned that there are 'partially completed runs'. Does that mean those experiments had not been finished before the submission of the paper, or what is meant by partially completed?

Correctness: Seems correct.

Clarity: A few things a are bit unclear in addition to the points mentioned under weaknesses: - What is meant by 'spatially-extended' (beginning of Sec 3)? - How was \sigma_p chosen? - It is mentioned that '80 epochs is sufficient for convergence'. How was this assessed? There is no mentioning of a validation set that was used to monitor training.

Relation to Prior Work: Discussion of related seems adequate. The experiments make several comparisons to state-of-the-art methods.

Reproducibility: No

Additional Feedback: For comparison, it may be good to also state the results of SegSort with the pre-trained ResNet-101 Experimental section feels rushed and could be improved by adding more insights and discussion of limitations and failure cases. I was a bit surprised that there is no supplemental material with further visual results. As a side note (and this has not impacted this review), I personally would have not chosen the image of a person with an alcohol bottle as the key image to illustrate the method. Surely, these public datasets have consent of people to be used (I assume) but I could imagine that person not be too happy to be featured that way. --- AFTER REBUTTAL: The author rebuttal is convincing with good clarifications and several new results. In the light of the rebuttal and the other reviews, I am upgrading my score and recommend acceptance.


Review 3

Summary and Contributions: === Post-rebuttal update === For full transparency, I copy my message from the discussion phase below: I was the most positive reviewer and would like to maintain my score. It looks like all reviewers (including myself) found the experimental section somewhat lacking, albeit along different dimensions. Nevertheless, the method is interesting by itself and the author response included several new experiments which partially addressed the weaknesses. Therefore I would argue to accept this paper. However, I would like to point out that, like R1, I also do not fully agree with the author's arguments on the evaluation protocol. === End of Post-rebuttal update === The paper proposes a new angle to self-supervised learning by employing hierarchical visual grouping. In particular, they use a hierarchical grouping algorithm (trained on only 500 images) and apply this to a new image, resulting in a hierarchy of regions. Given two pixels, they define a distance between them based on this hierarchy, which is turned into a probability of the pair belonging to the same 'visual primitive' (positive pair) or not (negative pair). They use these pairs in a contrastive feature learning framework. Note that their framework is pixel-based, rather then image-based as is the case in most self-supervised frameworks. They demonstrate decent improvements over previous work.

Strengths: - The paper uses perceptual grouping in self-supervised learning. I fully agree with the authors that perceptual grouping seems relevant to human vision, but is less popular in deep-learning based computer vision. I am happy to see a work that successfully exploits such classical ideas in a modern framework. - Results are quite reasonable and outperform the recent Momentum Contrast [19] - Paper is well written. - Figure 3 nicely shows the effect of cross-image negative sampling.

Weaknesses: The experimental section has a few problems: - L210 states that the authors use official code of [19,21,45], where they mostly preserve the default parameters. This is fine for the experiments on ImageNet in Table 1, which are convincing. However, I don't think this is fine for the rest of the table (when training on Pascal+COCO). Generally, hyperparameters make a huge difference, hence it is possible that the benefits presented here are mostly due to having found better hyperparameters for the method of the authors. - L201: "In experiments finetuning on PASCAL val". Is this a typo? Otherwise you train and test on the same 'val' set (see also caption Table 1). - Given the nature of the method, it would make sense to use segmentation as one of the tasks. It is expected that the proposed method would provide more improvements w.r.t. image-based self-supervised methods. This is a missed opportunity. - Why is there a "Semantic Segment Retrieval" task? This is a very uncommon task. Is this the same as the "Unsupervised Semantic Segmentation task in SegSort [21]? If yes, why are the results so much lower than in Table 3 of [21]? - Optimizing embeddings based on pixels rather than images intuitively uses more signal per image. This suggests that the method in this paper will be especially advantageous in the low-data self-supervision regime. The authors could verify this by running MoCo on PASCAL only.

Correctness: Yes, but see disclaimers on experiments above in Weaknesses.

Clarity: Yes.

Relation to Prior Work: Yes.

Reproducibility: Yes

Additional Feedback: The paper describes a nice idea for self-supervision based on perceptual grouping. I like the use of perceptual grouping, which intuitively feels important to human vision. However, I feel that the paper falls short of a very good paper (top 50% of accepted papers) because of the weaknesses in the experimental section. Please only address my weaknesses section in the author response. Smaller comments are below: - L32L Li et al. Citation missing. - While the current work uses an edge detector trained on 500 images, video could provide a way to make it completely unsupervised. See https://arxiv.org/abs/1511.04166 (CVPR 16). (was this the missing reference?). - The sampling procedure in L147 is unclear. Do you first sample positive and negative pixels using Eq. 2, 3, and then uniformly select 7 of those? Or do you randomly sample pixels, apply Eq 2, 3, and stop once the desired number of positive and negative pixels has been reached? Also, it looks like a single pixel j can serve both as positive and negative to i, right? - L169L PASCAL VOC does NOT only provide 1464 and 1449 images for train and val. This is an error also in [21]. PASCAL VOC provides all 10582 training images (L171). The original PASCAL VOC was about 20 object classes. In PASCAL context they added additional classes including 'stuff' classes.


Review 4

Summary and Contributions: The authors propose an approach to self-supervision that relies on the use of a contour detection algorithm trained on a small supervised set (BSDS, 500 images) to produce hierarchical segmentations of large unlabeled sets, which are then used to train a CNN from scratch by self-supervision. The authors use the hierarchical segmentation to derive a distance between pixel representations based on how many merges in the hierarchy would be required to place the two pixels within the same segment. The authors evaluate on a number of datasets and find that they outperform the baseline (Momentum contrast) by a large margin when the self-supervision dataset has a natural distribution (e.g., COCO) but only matches the baseline when it is highly curated (e.g., ImageNet). The main contribution of the proposed approach is the use of a "lower-level" trained algorithm to train a self-supervised algorithm for a "higher-level" task--in this case contour detection is considered a lower level task and segmentation is considered a higher level task.

Strengths: - the idea of self-supervision of higher-level tasks based on lower-level trained algorithms is interesting and intuitive, and it is practically of interest since higher-level tasks usually require more effort and data to annotate - the proposed algorithm learns embeddings at the pixel level rather than image level, which is more fine-grained - the proposed algorithm does not require moving-average encoders to enable large scale training, as the baseline (MoCo) does - the authors outperform the state of the art by a large margin in a few settings

Weaknesses: - I understand the theoretical desire to avoid using ImageNet labels in the contour detector, but the authors admit that ImageNet labels do improve contour detection. Why not have an experiment where ImageNet is used for the contour detector simply for the purpose of evaluating the impact of better contour detectors on self-supervision performance? - the authors only compare to SegSort [21] in the setting where the original approach is modified to use ResNet-50 with random training rather than ResNet-101 pre-trained on ImageNet as in [21]. While it's reasonable to compare in the modified no-pretraining setting, it is also informative to understand how the proposed approach compares to the approach as proposed in [21].

Correctness: Looks correct. It would be good to also show the original performance of techniques that were modified to compare on equal footing (e.g., [21]), for completeness.

Clarity: Yes, the paper is well written.

Relation to Prior Work: Yes.

Reproducibility: Yes

Additional Feedback: See weaknesses. POST REBUTTAL UPDATE: I was leaning toward acceptance even before the rebuttal, but now that I have read the rebuttal, the other reviews, and the discussion, I am convinced that the paper is worth accepting. The authors addressed my concerns, and I think there sufficient content to merit acceptance. I am upgrading my rating accordingly.

[Author Response · NeurIPS 2020]

We thank the reviewers for valuable feedback. Before addressing individual comments, we clarify common concerns.

**Evaluation Protocol:** The most ambitious aim of self-supervised learning is to create universal visual representations
that do not require end-to-end fine-tuning, but are instead portable to new tasks merely through additional of a shallow
auxiliary network. Moreover, "image-level" vs "pixel-level" training has no bearing on the validity of evaluating with
respect to this stricter protocol. Any method that uses a CNN learns more than just "image-level" representations; for
example, a CNN's internal activations can always be converted into per-pixel embeddings using hypercolumns (as we
do; see L189). While we agree that adding more fine-tuning results is important for expanding comparison to past work,
and will do so, we believe our more ambitious evaluation methodology is fundamental to driving progress in the field.

**Dataset Properties:** Progress in supervised object segmentation has been driven by a series of datasets with increasing
label complexity per scene, from PASCAL to COCO to LVIS[1]. It is reasonable to suspect a similar trend will hold for
self-supervised learning, in which case it will be essential to move away from methods such as [19,5,7], which build
pipelines dependent upon the single-object bias of the dataset (ImageNet). Corroborating this view are recent results[2]
suggesting even supervised ImageNet pretraining is of minimal value for learning visual tasks on complex scenes.

**Experiments:** As we are interested in learned embeddings that *universally* and *directly* fit a broad range of tasks
(*e.g.* instance tracking, segment search, semantic segmentation), we freeze the backbone in most experiments. We have
now also run the end-to-end fine-tuning experiment on COCO+VOC pretrained backbone, following the design in [19]
to replace DeepLabV3 head with two stacked convolution layers. Results are: ours $47.2$ vs MoCo $46.9$ mIOU. We use
a simpler siamese training architecture, instead of a momentum encoder and memory bank, yet achieve comparable
results to MoCo. With frozen pretrained backbone, we also run instance segmentation tasks on COCO2014, following
Mask RCNN in [19]. Results of ours vs MoCo: box AP $18.62$ vs $14.75$ and mask AP $17.82$ vs $14.31$.

Suggested by **R4**, we retrain our model on COCO+VOC with HED edges and achieve $49.9$ mIOU in above mentioned
end-to-end fine-tuning settings and $48.8$ (raised from $46.5$) mIOU in the frozen backbone experiment. Suggested by **R3**,
to demonstrate the efficiency of proposed pixel-wise optimization, we train MoCo on VOC only for the same number of
iterations (32K) and get $26.7$ mIOU (ours is $43.5$ mIOU) evaluated by freezing the backbone.

Figure 1: Visualization of success and failure cases denoted by mIOU on segment search task.

**R1 – COCO and VOC less curated?** LVIS[1] analyzes statistics of COCO images, finding many instances per-image
and a long-tailed category distribution, which supports our characterization. While Instagram-1B does not appear to be
publicly available, we worry that Instagram images might suffer from significant photographer bias.

**R2, R3, R4 – SegSort.** SegSort focuses on a different task: using contours to refine ImageNet pre-trained features,
producing semantic segmentation. Our task is to learn pixel-wise semantic-aware embeddings from scratch. To compare,
we evaluate SegSort with its backbone initialized from scratch. Performance in "semantic segment retrieval", the truly
unsupervised equivalent of SegSort's "unsupervised semantic segmentation" is this lower than their reported scores.

**R1, R2 – Partial ImageNet results.** We will update the final version to reflect the full 200 training epochs.

**R1, R2 – Sampling details.** We first sample regions, then a fixed number of pixels within chosen regions. Probability
should be summed over regions, *i.e.* $\sum_{R_b} P(j \in Pos(i)) = 1$. Pixels could be sampled as positive and negative at the
same time. Instead of a memory bank, we compute the representation from both original and augmented images.

**R1 – Distance in tree.** We compute $d_T(R_i, R_j)$ following Sec. 4.2 of [2], originally introduced in Sec. 2-4 of [1].

**R3 – Hyperparameters.** We pick $\sigma_p$ using limited search (0.2 - 1.0, spaced by 0.2). For MoCo training on COCO +
VOC, we also ran a limited hyperparameter search and did not see significant impact on results.

**Citations. R1:** We will add suggested papers. **R3:** Missed citation is indeed Li *et al.*, CVPR 2016.

**Misc. R1:** Table 1 row 9 should be 36.15. We will update Fig. 3, 4, 5 with best/worst cases (for Fig. 4, some examples
are shown above in Fig. 1). **R2:** 'Image' under 'Cross-Image Negative Sample' refers to treating the image-based
embedding as negative features. 'spatially-extended' means we preserve the spatial dimension in the embedding and
output a feature map. **R3:** We fine-tune on PASCAL $train\_aug$ and evaluate on PASCAL $val$.

## Footnotes

[1] A. Gupta, P. Dollar, R. Girshick. LVIS: A Dataset for Large Vocabulary Instance Segmentation. CVPR, 2019.

[2] K. He, R. Girshick, P. Dollar. Rethinking ImageNet Pre-training. ICCV, 2019.


[Meta-Review · NeurIPS 2020]

This is an interesting paper bringing a sober perspective on the recent self-supervised learning progress in ImageNet. It shows that there are still opportunities in going beyond image-level self-supervised tasks, achieving slightly better results than MOCO and other baselines on semantic segmentation, region retrieval and tracking, while simplifying some aspects (but requiring a pre-trained edge detector). All the reviewers agreed about acceptance and so do I. Please follow the reviewers suggestions for the camera ready. Definitely update the ImageNet numbers. Also worth making sure the semantic segmentation results are as much apples to apples as possible in terms of the heads put on top the ResNet-50 backbone.